# Lake and mire isolation data set for the estimation of post-glacial land uplift in Fennoscandia

Jari Pohjola[1], Jari Turunen[1], and Tarmo Lipping[1]

[1]Tampere University, Pohjoisranta 11 A, Pori, Finland

*Correspondence to:* Jari Pohjola (jari.pohjola@tuni.fi)

**Abstract.** Postglacial land uplift is a complex process related to the continental ice retreat that took place about 10,000 years ago and thus started the viscoelastic response of the Earth's crust to rebound back to its equilibrium state. To empirically model the land uplift process based on past behavior of shoreline displacement, data points of known spatial location, elevation and dating are needed. Such data can be obtained by studying the isolation of lakes and mires from the sea. Archaeological data on human settlements (i.e., human remains, fireplaces etc.) are also very useful as the settlements were indeed situated on dry land and were often located close to the coast. This information can be used to validate and update the postglacial land uplift model. In this paper, a collection of data underlying empirical land uplift modelling in Fennoscandia is presented. The data set is available at https://doi.pangaea.de/10.1594/PANGAEA.905352 (Pohjola et al., 2019).

## 1  Introduction

Holocene land uplift has been known to people living along the shores of the northern Baltic Sea for centuries. Land uplift is a consequence of the Weichselian stadial and the massive ice layer that covered the Northern Europe from the British Isles to the proximity of Ural mountains in Russia. Heavy ice load pressed the Earth's crust down during the Weichselian period and when the ice started to melt at the beginning of the post-Weichselian or the Holocene period, the crust began to rebound to its equilibrium state. During the Holocene land uplift, the area of the current Baltic Sea experienced several lake and sea phases before finally settling to the present form of the Baltic Sea. More detailed review of the history of the Baltic Sea and the changes in the sea level can be found in Tikkanen and Oksanen (2002), Björck (1995), Punning (1987) and Ojala et al. (2013).

The chronological order of the ice retreat phases is also important in land uplift modelling. The estimated time of withdrawal of the edge of the ice sheet from a certain location marks the phase when subsidence turned into land uplift at that particular location. Several studies are available considering the timing of ice retreat, e.g. Hughes et al. (2016) and Stroeven et al. (2016).

In empirical land uplift modelling, the most important sources of information are the ones describing the shoreline displacement. Ongoing land uplift can be monitored using precise GPS station networks and gravimetric measurements from satellites, described, for example, in Lidberg et al. (2010), Müller et al. (2012), Poutanen et al. (2010) and Timmen et al. (2004). Unfortunately, the GPS-based time series are relatively brief compared to other, less accurate data. The historical land uplift rate in Fennoscandia has been studied using information on lake isolation from the sea. Eronen et al. (2001) and Cato (1992) examined the isolation of several lakes using sediment samples taken from the bottom of the lakes. The samples were dated using the

[14]C radiocarbon dating method and the age and depth of the layer where saltwater algae changed into freshwater algae were determined. The resulting shoreline displacement curves show that the land uplift rate has not been steady and there have also been local variations in the uplift. The same technique has been used with the ponds and mires by dating the first organic layer on top of the inorganic clay layer. Generally, the organic matter accumulated on the bottom sediments of the basins indicates ice retreat. The basins collected to the data set are from the time period of the Ancylus Lake (9500-8000 BP (Björck, 1995; Tikkanen and Oksanen, 2002)), so it can be assumed that the ice had melted before the accumulation of organic matter. Although the timing might not be exact, it is an indication of the isolation. In addition, there are geological interpretations of the retreat of the Ancylus Lake and the Baltic Ice Lake drawn from mire studies, e.g. (Eronen et al., 2001; Mäkilä et al., 2013).

Archaeological findings are useful in cases where the sea level can be checked against the location and age of human settlements. In Fennoscandia, it can be safely assumed that people during the Holocene lived on dry land and, therefore, the location and elevation of these settlements set an upper bound for the sea level at the time the settlements are dated to.

In this paper we present a data set underlying our efforts in developing empirical land uplift models (Pohjola et al., 2014). The data set consists of lake isolation data, data based on the initial layers of mire development as well as relevant archaeological findings. In the next chapters we describe the data and the data cleaning procedure and present the data collectors and the owners of the data repositories.

## 2 The data sets

### 2.1 Data description

The data set presented in this paper is a collection of 2335 data points and it can be divided into two subsets: lake/mire isolation data (1086 data points) and archaeological data (1249 data points). Lake/mire isolation data are represented by red circles and archaeological data by black triangles in Figure 1. Each data point has a spatial location, elevation and age information that has been obtained using the [14]C dating method. The [14]C dates in the data sets are uncalibrated so they need to be calibrated in order to convert them to calendar ages. In most cases, the uncertainties in the [14]C dating method are also taken into account, but the user of this data set must be aware of the uncertainties in spatial coordinates and elevation values (explained in more detail in the 'Data Handling' section). In the following sections, the sources of the two data subsets are presented. An example of the data from both data sets can be seen in Figure 2.

### 2.2 Lake and mire isolation data set

The lake/mire isolation data forms the main data set used in the modelling. The basic idea behind these type of data is the determination of the freshwater/saltwater ecosystem boundary from a core drilled sample from a lake or mire bottom sediment. The isolation is defined by the transition from marine or brackish water algae to fresh-water algae. These kinds of samples have been collected especially in the area of Finland and Sweden. In Finland, the main data contributors have been Matti Eronen and Gunnar Glückert (Eronen et al., 2001). Arto Vuorela collected additional data points and included the data set of Matti Eronen

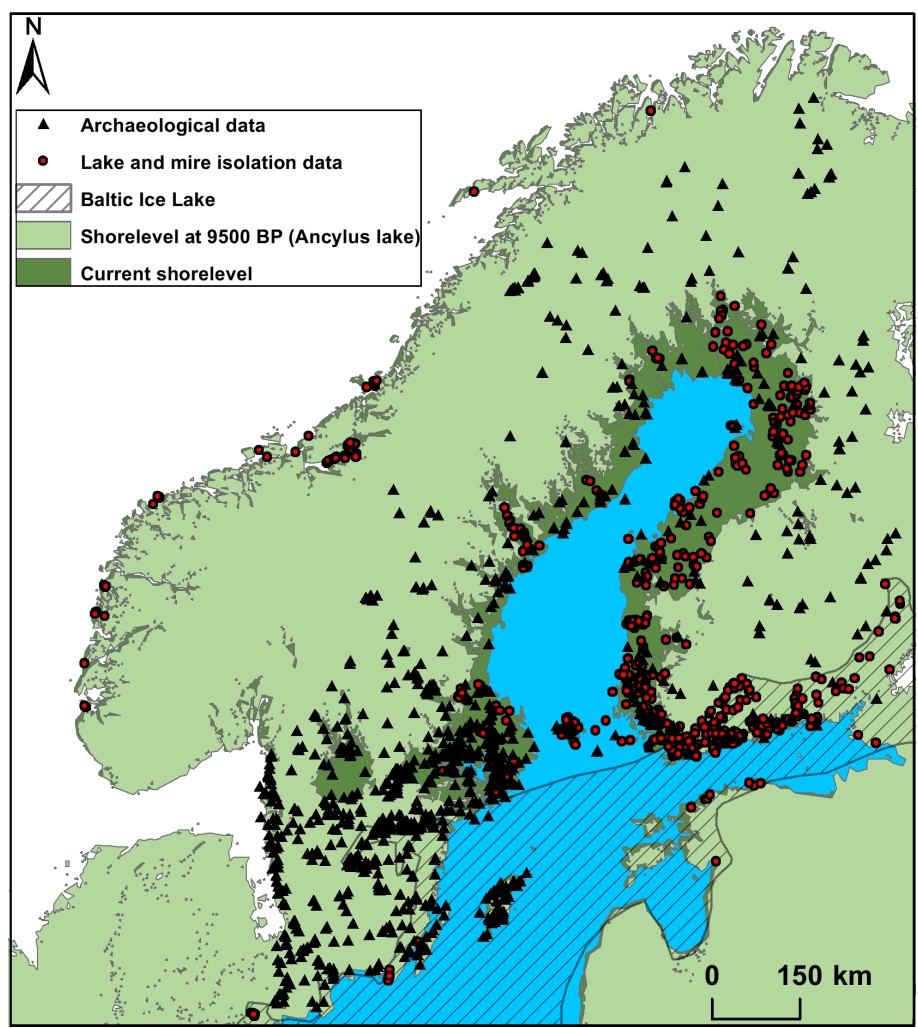

**Figure 1.** Spatial location of source data points. Maximum extent of the Ancylus lake phase is indicated in dark green color and the Baltic Ice Lake area is marked by a hatched area. *Data points: (Pohjola et al., 2019). Background map: Esri. Historical shorelines: Geological Survey of Finland.*

and Gunnar Glückert into his own data set, published in (Vuorela et al., 2009). Mäkilä et al. (2013) present a collection of mire evolution and isolation datings in Finland. The collection of data points presented in this paper is sampled selectively from these sources and complemented with data points gathered from other scientific sources. Pond and mire data sets are based on core drillings and in some cases there are large age discrepancies between samples that are close to each other in the core so

5   the interpretation of the results is important. In addition, in some rare cases the datings do not fully correspond to the depth of the dated sample along the core. An example of this kind age reversal are the samples belonging to the Nälköönsuo series (Jungner, 1979). The younger sample HEL-103 has been taken from a deeper layer than the older sample HEL-104. These kind

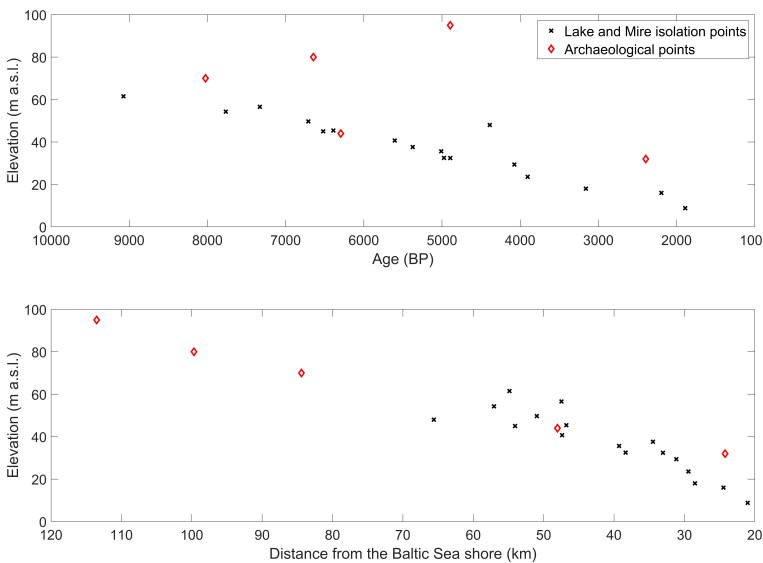

**Figure 2.** A sample of the data set containing data points along the line from (61° 0' 0,0", 21° 6' 0,0") to (61° 0' 0,0", 24° 0' 0,0") in WGS84 coordinate system. The distribution of the data points in the elevation – age and elevation – distance domains is presented in the upper and lower panel, respectively. Black cross markers denote the lake/mire isolation data and red diamond markers the archaeological data. *Background map: Esri.*

of discrepancies can result from e.g. redeposited older material from outside the catchment or roots that were inadvertently dated giving ages that are younger than than the layer above. In these cases the oldest sample was considered even if it is not the deepest one near the clay sediment in the drilling core. Some Norwegian shoreline points experienced several oscillatory rebound and retreat phases. The same issue can be seen in Finland and Sweden, where the same spatial location may have shore markings of the Baltic Ice Lake, the Litorina Sea and possibly the Ancylus Lake. Fortunately, in most cases the elevation
5   is different and this allows to use all the available elevation/dating information of the point in question.

## 2.3   Archaeological data set

There are plenty of archaeological data available, but our main focus is to acquire samples that have an existing $^{14}$C dating. For example, if there were coal remains from different time periods in dwelling site fireplaces, the oldest dated sample was included in the data set. The oldest sample is also an evidence of the site in question being above the water level at that time. However,
10   there are issues of which the user of this data set must be aware. For example in Kolmhaara, Eura in southern Satakunta, Finland, the radiocarbon dated burial remains seem to be older than expected from the radiocarbon datings. Kolmhaara is a famous place, because it contains human remains (graves, fireplaces etc.) from a time period stretching over thousands of years. The site has been located near the Baltic Sea shoreline. However, when using shoreline modelling, these burials appear to be

several meters below the sea level at the time the remains are dated to. Either the shoreline displacement model is incorrect or the 'Marine reservoir effect' (Dettman et al., 2015; Reimer and Reimer, 2006) has had an influence on the samples. Also, the 'Old wood effect' (Olsen et al., 2013) might have had affected the samples. It is concluded in Olsen et al. (2013) that the dating of cremated human bone could have an inaccuracy of approximately 50-100 $^{14}$C years. In addition, the freshwater reservoir effect (Philippsen, 2013) is one aspect that could be considered with the time and elevation discrepancies.

Main sources of the archaeological findings are the Finnish Heritage Agency archives (http://www.kyppi.fi) and the Swedish National Heritage Board archives (http://www.raa.se). $^{14}$CARHU database consists of the radiocarbon data collected by Helsinki University (Junno et al., 2015). More detailed references are included in the data sets.

## 3   Data handling and organisation

Various different coordinate systems were used in the original data sets. The data were reprojected to the WGS84 coordinate
system and the elevation is presented as meters above the sea level corresponding to the average sea level datum. Some of the archaeological point locations were described verbally like *"half a kilometer west from the main building..."* and some of the site locations were drawn manually illustrating the significant landmarks in the area. All these data points were checked using the National Land Survey of Finland (NLS) maps as some of the archaeological sites are included in the NLS base maps. If they were not included in the base maps, the best estimates, based on the verbal clues and possible hand drawings, were
determined using the NLS maps. In some cases the elevation information was also checked from the NLS base maps or NLS digital elevation maps and refined if possible. In the cases where elevation was known, it was possible to check the location using elevation information. The maximum error in the spatial location of the data points can be estimated to be about 100 meters, however, as a single data point usually represents a larger area of several kilometers' radius when estimating the land uplift behavior, the exact location of the data points is not critical. As an example, a part of this data set was used in the study
by Pohjola et al. (2018).

The vertical error in the lake/mire isolation and archaeological data sets is assumed to be ±0.5 m. Defining the altitude is sometimes very challenging, especially in villages situated at hill slopes. In these cases the lowest altitude is used and it is assumed that the lowest point of the village has existed during the oldest period. This is somewhat questionable, because the village might have expanded during the centuries around the oldest settlement. As said before, the archaeological data set is
used only for validation purposes when modelling land uplift.

In all cases concerning both the archaeological and the lake/mire isolation data, latitude (WGS84), longitude (WGS84), altitude (meters above sea level), $^{14}$C age (uncalibrated years BP), $^{14}$C error (years), the name of the place, the reference ($^{14}$C laboratory identification, if available), URL of the database (if available), dated material (if available) and possible additional information are provided. The reference field will contain the $^{14}$C laboratory report or a link to a database. In some cases the
samples are collected as a side product of other work and in these cases the laboratory number is the link to digital site report (in Finnish) or some commonly referred article, where the finding is mentioned.

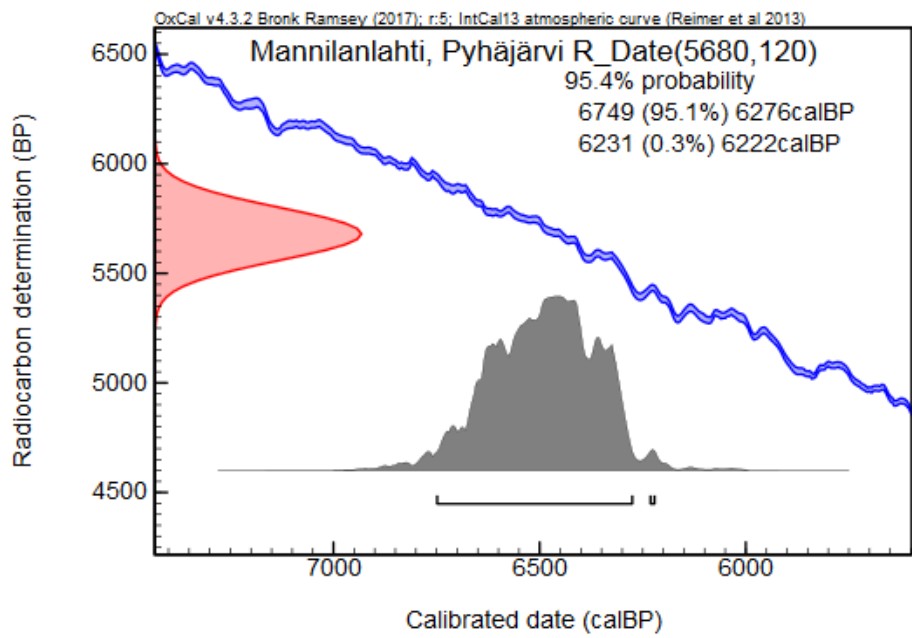

**Figure 3.** The calibration curve and the error distribution from OxCal program (Bronk Ramsey, 2009).

In earlier studies, (e.g. Pohjola et al. (2014, 2018)) the calibration for the $^{14}$C radiocarbon dated data points was done using the Oxcal program (Bronk Ramsey, 2009) containing Oxford radiocarbon acceleration unit calibration curve (IntCal 13) (Reimer et al., 2013). An example of the calibration curve is presented in Figure 3.

## 4    Conclusions

This data set is intended to be used for modelling the postglacial land uplift in Fennoscandia between 12,500 and 0 years BP. The collected data set in this paper covers the coastal area of Finland and Sweden reasonably well, especially the extent of the
5   Ancylus Lake. There is always room for improvement, because lake and mire data points are missing from e.g. Hälsingland and Västerbotten areas in Sweden. However, the archaeological data points cover sufficiently almost all of the Swedish - Finnish coastal area. Also on the Estonian side, more archaeological and geological data are needed.

The data set introduced in this paper can also be used to estimate the flooding risk due to the rising seawater. In Bothnian Bay the land uplift estimates (6-9 mm / year) (Poutanen et al., 2010) are higher than the predicted seawater rise (2.5 - 5.5
10   mm / year) (Church et al., 2013). In contrast, the land uplift in the Gulf of Finland is 0-5 mm / year (Poutanen et al., 2010). Therefore, future housing development planning in the northern Baltic Sea coastal areas can benefit from this data set as a tool to assess the flooding probability.

# 5 Data availability

Both datasets are available at https://doi.pangaea.de/10.1594/PANGAEA.905352 (Pohjola et al., 2019).

*Author contributions.* J.P. and J.T. performed the data collection and handling. The manuscript was written by J.P., J.T. and T.L. in collaboration.

*Competing interests.* The authors declare that they have no conflict of interest.

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
