# Peer review of "Lake and mire isolation data set for the estimation of post-glacial land uplift in Fennoscandia"

_Earth System Science Data, 2019_

## Referee Comment (RC1) · Anonymous Referee #1 · 22 Nov 2019

General Comments: This manuscript provides a lake and mire isolation data set for the estination of post-glacial land uplift,the data sets focuses on the lake and mire isolation data set and archaeological data set. although the data set can forms the main data set used in the modeling, the manuscript should be not publicated the ESSD, I have three points to reject it. (1) the manuscript only collect the data, it is short of originality of the study. (2) the dataset is at small spatial extent, they will not attract the wide readers of the world. (3) the authors should added the detailed the description of the method in producing the data set

---

## Referee Comment (RC2) · Anonymous Referee #2 · 19 Dec 2019

The article 'Lake and mire isolation dataset for the estimation of post-glacial land uplift in Fennoscandia' of Pohjola et al. presents a collection of data, drawn from existing, both archaeological and palaeoenvironmental sources. It has been made available on the PANGAEA database. The data covers the complete Holocene and provides information about the ages of the earliest radiocarbon dates from mires and lakes, supposed to be representing their earliest stages after being isolated from the Gulf of Bothnia. Combined with spatial information (location and elevation), this is useful to build or validate and optimise land uplift models for Fennoscandia. Some potential pitfalls or deficiencies in the use of this data are pointed out. However, certain parts need some more critical discussion while others need clarification in order not to confuse or misguide readers and potential data users. Most of it concerns radiocarbon dating.

[Figure]

Furthermore, the dataset uploaded to PANGAEA could benefit from certain additions, especially for the case that inconsistent results need to be evaluated critically. It would also get more interesting for disciplines apart from postglacial uplift modelling. The conclusion seems to devaluate the dataset, as it suggests that it should not be published yet by saying that the search for additional (already existing) data is ongoing. All relevant studies that are already published should already be in the PANGAEA data of this manuscript. I suggest revision and a re-evaluation. Detailed comments can be found below.

Detailed comments:

l. 20: Add more references if there are any or put an 'e.g.' before the two given citations or mention that these two are the most important (if so).

l. 21: Not a complete sentence ('the most important source of information is that describing the shoreline displacement'). Please rephrase to make clear what was intended to be said.

ll. 5-7: This needs to be discussed in further detail. Consequently, ages that are from the first organic layers of ponds or mires are rather indicating the age of the ice retreat but cannot safely be used to infer uplift or isolation from marine influence.

l. 27: define the kind of laboratory analysis or rephrase. For example, simply to: 'The isolation is defined by the transition from marine or brackish water algae to fresh-water algae.'

ll. 1-2: Bog and mire? Above, it was pond and mire. As a bog is also a mire, but a mire is not always a bog, please correct. See also page 2, l. 31.

l. 4: Having information on the material that was sampled in both PANGAEA data files would be very good. It would allow the reader to assess the reliability of the radiocarbon age. Depending on the type of macrofossil or sediment dated, the discrepancies can be large.

Figure 2: Why is the term height used, instead of elevation? The unit of Age is given in cal year instead of years BP, as in the text. Be consistent throughout manuscript and dataset.

ll. 7-8: Do you mean older than expected? Something in this sentence is wrong: 'the radiocarbon dated burial remains seem to be younger than expected from the radiocarbon datings.' Expected from which datings? Did you want to say that the burial remains should be younger than the radiocarbon dating suggests?

l. 9: Do you mean 'from a time period stretching over thousands of years' or 'from thousands of years ago'? Consider revising to be more precise.

ll. 10-12: What about an old wood effect of the dated material? Depending on what type of archaeological material was dated, the marine reservoir effect is unlikely to have altered the sample. What about the possibility that bones of humans were dated, which were eating a lot of fish or molluscs. And: also, freshwater lakes have varying reservoir effects (e.g. Philippsen 2013). Again, the type of material dated (bulk sediment, plant macrofossils, wood, bones, etc.) would be valuable to add directly to the data files provided on PANGAEA.

ll. 19-22: The term 14C age is normally used for uncalibrated radiocarbon ages (also concerns the 14C-error). In addition, the terminology is different in the .tab data files, where only Age and error are used. Furthermore, the ages given in the .tab file appear to be uncalibrated (for example Hel-146 in archaeological data). It is not only confusing

but also dangerous if the data is used wrongly by users who believe to have calibrated ages at hand. This needs to be resolved by clear terminology in manuscript and both data files, and maybe with a comment for readers who are less experienced with radiocarbon dating. It needs to be clear if the age is calibrated or not. Right now, the text suggests that calibrated ages are provided, which is not the case in the data.

ll. 21-22: Links/URLs to a database or report in lake/mire data file are not existing. Why not?

ll. 25-26.: The Bronk Ramsey citation should be put behind 'Oxcal program' in l. 24. The IntCal 13 calibration curve should be cited properly with Reimer et al. (2013). The Reimer reference also appears on Figure 3, so it needs to be in the full references anyway.

ll. 25ff.: Does this paragraph relate to the data handling to produce for example Figure 2? It comes a bit out of nowhere as long as the previous lines are saying, that the data is already calibrated. As the radiocarbon data in the data files is not calibrated, consider introducing this paragraph in a different way to put it more into the context of how to handle the data. Right now it says, that "the calibration was done using [. . .]", but where was it done then?

ll. 30-31: 'certain areas' and deferring to Fig. 1 is a bit vague. This should have been at least briefly discussed earlier in the text. For example, it is said in the beginning, that the lake/mire isolation data is the most important for modelling, but the data points are concentrated mostly on the eastern coast of the Gulf of Bothnia.

Figure 3.: In the example for a calibration, the scale is set to calBC, which is not mentioned before. See comment above concerning the consistent use of units.

Comments on the files uploaded to PANGAEA ('Fennoscandia_lake_mire_isolation.tab' and 'Fennoscandia_archaeological_data.tab')

- The archaeological data provides URLs to the original radiocarbon data. Unfortunately, this is not the case for the lake/mire isolation data. Why are no URLs or further information provided?

- Referring to the problem of using pond/mire data (page 2, ll. 5-7), how can the data user distinguish between lake radiocarbon data and pond/mire radiocarbon data?

- The whole dataset would greatly benefit from adding the material that was used for radiocarbon dating. By this, the reliability of the ages could be assessed better. Furthermore, disciplines aside from uplift modelling would get attracted to the data collection.

Technical comments:

l. 3: its instead of it's

l.14: see previous comment

ll. 16-17: Correct citation style '. . .can be found in Tikkanen and Oksanen (2002), Björck (1995), Punning (1987) and Ojala et al. (2013).

l. 22: 'Nowadays, land uplift. . .' or 'Ongoing land uplift...' or 'Today's land uplift..'

l. 24: citation style

l. 25: citation style ('Eronen et al. (2001) and Cato (1992) examined the isolation of several lakes. . .')

l. 5: 'on top of'

l. 19: remove 'timing', put 'age' information

l. 28: 'In Finland, the main data. . ..'

ll. 30-31: Citation style (brackets). Consider rephrasing:' Mäkila et al. (2013) present a collection of . . .'

Figure 1: consider highlighting the Baltic Ice Lake better, as the contrast of the mild blue to the background is partly not high enough. Also think about the graphic being printed in black and white.

l. 1: I would use British English (organisation), as you are also using 'archaeology' and not 'archeology'.

ll. 20-21: Something is wrong with the structure and the brackets here:'. . .the name of the place, the reference (14C) Laboratory Identification, if available),. . .'

l. 32: citation style (see above)

---

## Author Comment (AC1) · 27 Jan 2020

We would like to thank the two Anonymous Referees for their comments that improved the quality of the manuscript. The detailed responses to each comment can be found further on in this document.

Anonymous Referee #1

General Comments: This manuscript provides a lake and mire isolation data set for the estination of post-glacial land uplift,the data sets focuses on the lake and mire isolation data set and archaeological data set. although the data set can forms the main data set used in the modeling, the manuscript should be not publicated the ESSD, I have three points to reject it. (1) the manuscript only collect the data, it is short of originality

of the study. (2) the dataset is at small spatial extent, they will not attract the wide readers of the world. (3) the authors should added the detailed the description of the method in producing the data set

- We thank Referee #1 for the comments. Here are detailed responses for the comments:

(1) This is true, but the combination and unification of two different data sets including lot of data from different sources producing a usable data set for the estimation of the land uplift in Scandinavia makes it worthwhile to present the manuscript and the data in our opinion.

(2) The land uplift is a phenomenon that is notably present at only few locations in earth and the studies concerning it have been mainly done in Scandinavia so this makes the study focused around the Baltic Sea area.

(3) Some details of the method in producing the data set have been added to chapters 1 (page 2, lines 10-20) and 3 (page 6, lines 4-6).

Anonymous Referee #2

The article 'Lake and mire isolation dataset for the estimation of post-glacial land uplift in Fennoscandia' of Pohjola et al. presents a collection of data, drawn from existing, both archaeological and palaeoenvironmental sources. It has been made available on the PANGAEA database. The data covers the complete Holocene and provides information about the ages of the earliest radiocarbon dates from mires and lakes, supposed to be representing their earliest stages after being isolated from the Gulf of Bothnia. Combined with spatial information (location and elevation), this is useful to build or validate and optimise land uplift models for Fennoscandia. Some potential pitfalls or deficiencies in the use of this data are pointed out. However, certain parts need some more critical discussion while others need clarification in order not to confuse or misguide readers and potential data users. Most of it concerns radiocarbon dating.

Furthermore, the dataset uploaded to PANGAEA could benefit from certain additions, especially for the case that inconsistent results need to be evaluated critically. It would also get more interesting for disciplines apart from postglacial uplift modelling. The conclusion seems to devaluate the dataset, as it suggests that it should not be published yet by saying that the search for additional (already existing) data is ongoing. All relevant studies that are already published should already be in the PANGAEA data of this manuscript. I suggest revision and a re-evaluation. Detailed comments can be found below.

-We thank Referee #2 for the constructive comments. They have improved the quality of the manuscript greatly. Also, the data set uploaded to PANGAEA will be updated based on the comments. The conclusion has been rewritten to give more basis for the study. Here are detailed responses for the comments.

Detailed comments:

l. 20: Add more references if there are any or put an 'e.g.' before the two given citations or mention that these two are the most important (if so).

- e.g. has been put before the citations and the citation style has been corrected

l. 21: Not a complete sentence ('the most important source of information is that describing the shoreline displacement'). Please rephrase to make clear what was intended to be said.

- Corrected as 'the most important sources of information are the ones describing the shoreline displacement'

Page 2 ll. 5-7: This needs to be discussed in further detail. Consequently, ages that are from the first organic layers of ponds or mires are rather indicating the age of the ice retreat but cannot safely be used to infer uplift or isolation from marine influence.
- This has been discussed in more detail: 'Generally, the organic matter accumulated to the bottom sediments of the basins indicates ice retreat. The basins collected to the data set are from the time period of the Ancylus Lake, so it can be assumed that the ice had melted before the accumulation of organic matter. Although the timing might not be exact, it is an indication of the isolation contact. In addition, there are geological interpretations of the retreat of the Ancylus Lake and the Baltic Ice Lake in mire studies and references'.

l. 27: define the kind of laboratory analysis or rephrase. For example, simply to: 'The isolation is defined by the transition from marine or brackish water algae to fresh-water algae.'

- Corrected as suggested

Page 3 ll. 1-2: Bog and mire? Above, it was pond and mire. As a bog is also a mire, but a mire is not always a bog, please correct. See also page 2, l. 31.

- For clarification, the manuscript has been modified to use lakes, ponds and mires throughout the text.

l. 4: Having information on the material that was sampled in both PANGAEA data files would be very good. It would allow the reader to assess the reliability of the radiocarbon age. Depending on the type of macrofossil or sediment dated, the discrepancies can be large.

- A column for the dated material will be added to both data files.

Figure 2: Why is the term height used, instead of elevation? The unit of Age is given in cal year instead of years BP, as in the text. Be consistent throughout manuscript and dataset.

- The figure has been corrected with term elevation instead of height and the unit of

age has been changed to years BP

ll. 7-8: Do you mean older than expected? Something in this sentence is wrong: 'the radiocarbon dated burial remains seem to be younger than expected from the radiocarbon datings.' Expected from which datings? Did you want to say that the burial remains should be younger than the radiocarbon dating suggests?

- Yes, older than expected was meant and it has been corrected.

l. 9: Do you mean 'from a time period stretching over thousands of years' or 'from thousands of years ago'? Consider revising to be more precise.

- Corrected as 'from a time period stretching over thousands of years'

ll. 10-12: What about an old wood effect of the dated material? Depending on what type of archaeological material was dated, the marine reservoir effect is unlikely to have altered the sample. What about the possibility that bones of humans were dated, which were eating a lot of fish or molluscs. And: also, freshwater lakes have varying reservoir effects (e.g. Philippsen 2013). Again, the type of material dated (bulk sediment, plant macrofossils, wood, bones, etc.) would be valuable to add directly to the data files provided on PANGAEA.

- The old wood effect and the freshwater reservoir effect on radiocarbon dated samples have now been discussed: 'Also, the 'Old wood effect' (Olsen et al., 2013) might have had affected the samples. It is concluded in Olsen et al. (2013) that the dating of cremated human bone could have an inaccuracy of approximately 50-100 14C years. In addition, the freshwater reservoir effect (Philippsen, 2013) is one aspect that could be considered with the time and elevation discrepancies.' The material issue with the data files was discussed in a previous response.

ll. 19-22: The term 14C age is normally used for uncalibrated radiocarbon ages (also concerns the 14C-error). In addition, the terminology is different in the .tab data files,

where only Age and error are used. Furthermore, the ages given in the .tab file appear to be uncalibrated (for example Hel-146 in archaeological data). It is not only confusing but also dangerous if the data is used wrongly by users who believe to have calibrated ages at hand. This needs to be resolved by clear terminology in manuscript and both data files, and maybe with a comment for readers who are less experienced with radiocarbon dating. It needs to be clear if the age is calibrated or not. Right now, the text suggests that calibrated ages are provided, which is not the case in the data.

- This has been corrected in manuscript, the ages given are indeed uncalibrated. The column headers of the data-files will also be corrected.

ll. 21-22: Links/URLs to a database or report in lake/mire data file are not existing. Why not?

- URLs have been added where it has been possible

ll. 25-26.: The Bronk Ramsey citation should be put behind 'Oxcal program' in l. 24. The IntCal 13 calibration curve should be cited properly with Reimer et al. (2013). The Reimer reference also appears on Figure 3, so it needs to be in the full references anyway.

- The Bronk Ramsey citation has been put to its appropriate place and the IntCal13 reference (Reimer et al. 2013) has been added to the reference list.

ll. 25ff.: Does this paragraph relate to the data handling to produce for example Figure 2? It comes a bit out of nowhere as long as the previous lines are saying, that the data is already calibrated. As the radiocarbon data in the data files is not calibrated, consider introducing this paragraph in a different way to put it more into the context of how to handle the data. Right now it says, that "the calibration was done using [. . .]", but where was it done then?

- This has been clarified linking this paper to our previous works.

ll. 30-31: 'certain areas' and deferring to Fig. 1 is a bit vague. This should have been

at least briefly discussed earlier in the text. For example, it is said in the beginning, that the lake/mire isolation data is the most important for modelling, but the data points are concentrated mostly on the eastern coast of the Gulf of Bothnia.

- This has been written in more detail: 'The collected data set in this paper covers the coastal area of Finland and Sweden reasonably well, especially the extent of the Ancylus Lake. There is always room for improvement, because lake and mire data points are missing from e.g. Hälsingland and Västerbotten areas in Sweden. However, the archaeological data points cover sufficiently almost all of the Swedish - Finnish coastal area. Also in the Estonian side, more archaeological and geological data are needed.'

Page 6 Figure 3.: In the example for a calibration, the scale is set to calBC, which is not mentioned before. See comment above concerning the consistent use of units.

- The scale in the figure has been changed to years BP

Comments on the files uploaded to PANGAEA ('Fennoscandia_lake_mire_isolation.tab' and 'Fennoscandia_archaeological_data.tab')

- The archaeological data provides URLs to the original radiocarbon data. Unfortunately, this is not the case for the lake/mire isolation data. Why are no URLs or further information provided?

- URLs will be added where it is possible.

- Referring to the problem of using pond/mire data (page 2, ll. 5-7), how can the data user distinguish between lake radiocarbon data and pond/mire radiocarbon data?

- A column will be added to the data file to identify the basin type.

- The whole dataset would greatly benefit from adding the material that was used for radiocarbon dating. By this, the reliability of the ages could be assessed better. Furthermore, disciplines aside from uplift modelling would get attracted to the data collection.

- The material will be added to the data files.

Technical comments:

l. 3: its instead of it's

l.14: see previous comment

- Both were corrected as suggested

ll. 16-17: Correct citation style '. . .can be found in Tikkanen and Oksanen (2002), Björck (1995), Punning (1987) and Ojala et al. (2013).

- The citation style has been corrected

l. 22: 'Nowadays, land uplift. . .' or 'Ongoing land uplift...' or 'Today's land uplift..'

- Nowadays land uplift was changed to Ongoing land uplift

l. 24: citation style

- The citation style has been corrected

l. 25: citation style ('Eronen et al. (2001) and Cato (1992) examined the isolation of several lakes. . .')

- The citation style and the sentence have been corrected as suggested

l. 5: 'on top of'

- corrected as suggested

l. 19: remove 'timing', put 'age' information

- timing changed to age

l. 28: 'In Finland, the main data. . ..'

- corrected as suggested

ll. 30-31: Citation style (brackets). Consider rephrasing:' Mäkila et al. (2013) present a collection of . . .'

- corrected as suggested and the sentence has been rephrased

Figure 1: consider highlighting the Baltic Ice Lake better, as the contrast of the mild blue to the background is partly not high enough. Also think about the graphic being printed in black and white.

- The Baltic Ice Lake has been highlighted better, the figure has also been made more suitable for black and white print.

l. 1: I would use British English (organisation), as you are also using 'archaeology' and not 'archeology'.

- organization has been changed to organisation

ll. 20-21: Something is wrong with the structure and the brackets here:'. . .the name of the place, the reference (14C) Laboratory Identïfication, if available),. . .'

- the extra ) has been removed

l. 32: citation style (see above)

- The citation style has been corrected

Please also note the supplement to this comment:

https://www.earth-syst-sci-data-discuss.net/essd-2019-165/essd-2019-165-AC1-supplement.pdf

---

## Referee Report (RR1)

Dear Pohjola et al.,

Thank you for reacting to the comments of my first review by incorporating the suggested changes into the written part and figures of your manuscript. Beside a few minor comments (see below), the revision with the implemented corrections, additional information and discussion has improved its quality and readability.

Having checked the two datasets/files with the archaeological and mire/lake isolation data accessible on PANGAEA, I was not able to find changes or additional data yet. This concerns the requests for:

➔ changing of headers (age unit terminology, see first reviewer comment, it needs to be clear that the ages in the data files are uncalibrated)
➔ providing a separate column for the type of dated material (wood, bone, bulk peat, macrofossils, charcoal, etc.) in both files
➔ providing a separate column for the dated archive/basin type (lake, mire or pond)
➔ adding URLs to the radiocarbon reports of the lake/mire isolation data, where available (like in the archaeological data file).

Although there is a lot of dry work involved, I strongly suggest adding this information to raise the quality of the data and increase the impact of this publication. If changes were already made and the changes are just not visible because the files are in review on the PANGAEA website, just ascertain again, that the adjustments were correct.

Please find a few minor comments and technical issues below. Please excuse me that I might have overseen some of them in the first review:

l. 17: Instead of 'timing', the term 'chronological order' should be used.

l. 21: comma after satellites

l. 4: 'on the bottom sediments' instead of 'in'

l. 5: to the 'time period of Ancylus Lake', the time period should be given in numbers, with a reference.

l. 6: isolation and contact contradict each other. You could remove 'contact'.

ll. 7-8: '…in mire studies and references…', technically, these mire studies are among the references. You can write that there are interpretations of the retreat drawn from mire studies and then put a few literature references behind, which are specifically evidencing it. In other words, if you already use 'references', then also name a few.

l. 10: comma after 'In Fennoscandia'

l. 13: 'data based on expert judgement on the sediment layers of mires…' sounds weird, as if it is unclear what "expert judgement" is or who these experts are. It would sound more serious if you just write: 'data based on the initial layers of mire development as well as …'

ll. 20-21: put 'method' behind $^{14}$C.

'Each data point has a spatial location, elevation and age information that is preferably obtained using the 14C dating and calibration.' Because the chapter is called data description, the sentence suggests or may be misinterpreted, that the ages provided in the files are calibrated already. It should be made clear that the data user still needs to calibrate the 14C-dates. The (raw) 14C-dates are already there, but the calibration still needs to be done, which is not differentiated here yet.

l. 3: It is not completely clear what is meant by the statement that 'in some cases the layers of different datings are close to one another in the core.' Does it mean that there are big age discrepancies between samples that are almost in the same layer/depth or very close to each other?

ll. 3-4: You could give some hint to the reader for possible reasons for these age reversals. For example, roots that were accidentally dated and give ages that are less old than a layer above, redeposited older material from outside the catchment…. Possibly with a few references. At least, remove the 'for some unknown reasons'. Otherwise, it may sound slightly unprofessional.

In Figure 2, the figure caption still says height instead of elevation. Consider using m a.s.l. (above sea level).

ll. 20-21: Again the pointer to the two datasets and the need to apply the changes concerning terminology, units and additional columns on sample type, links to the reports where missing, etc. (see first review).

l. 25: Put citation/syntax in order/rephrase to: 'In earlier studies (e.g. Pohjola et al., 2014, 2018), the calibration…..'

Page 6 (mind that the line numbers are not displayed correctly)

l. 33: on the Estonian side

ll. 1-5: To streamline the conclusion slightly, consider ending the sentence after the Church et al. reference and then: 'In contrast, the land uplift in the Gulf of Finland is 0-5 mm / year (Poutanen et al., 2010). Therefore, future housing development planning in the northern Baltic Sea coastal areas can benefit from this data set as a tool to assess the flooding probability.'

Page 8 (References)

ll. 2-3: DOI is missing for this reference.

All pages:

'modeling' is American English. Change to British 'modelling'. Check for consistency throughout the manuscript.

Page 1, lines 7, 17, 20, Page 2, l. 26., Page 4, l. 10, Page 5, l. 18

---

## Editor Decision (ED1)

Dear Jan Pohjola et al,

many thanks for addressing all change requests by the reviewers. Before finally accepting the manuscript, I would like to address some very minor technical corrections for the manuscriot and the dataset:

Data:

1. Table: Fennoscandia_lake_mire_isolation.tab, rows 261-266 (Hedenström. Early Holocene shore displacement in eastern Svealand, Sweden): The provided link to the pdf (https://pubs.sub.su.se/5454.pdf) is not accessible from outside Sweden. This is ok as most of the data are in the table. However, it would be helpful to add a note in the data table saying "not accessible outside Sweden) and preferably changing the link from "https://pubs.sub.su.se/5454.pdf" to "http://su.diva-portal.org/smash/record.jsf?pid=diva2%3A1094181&dswid=863". The latter represents the landing page of this work and includes the information about the access restriction.

2. The DOI of the data are not registered. Please ask PANGAEA to register the DOI after addressing my first point. After the DOI is registered, please change all https://doi.pangaea.de links in the paper to https://doi.org (in the abstract, data availability statement and references)

Paper:

- Please address the changes in the DOI link to the data as mentioned above
- Page 3, line 6:  please change "An example of this kind age reversal..." to "An example for this **kind** of age reversal..."
-
- Page 4, first paragraph, second line: please change "younger than than" to "younger than"

Many thanks and best regards

Kirsten Elger

---

## Author Response (AR2)

Dear Pohjola et al.,

Thank you for reacting to the comments of my first review by incorporating the suggested changes into the written part and figures of your manuscript. Beside a few minor comments (see below), the revision with the implemented corrections, additional information and discussion has improved its quality and readability.

Having checked the two datasets/files with the archaeological and mire/lake isolation data accessible

on PANGAEA, I was not able to find changes or additional data yet. This concerns the requests for:

▢ changing of headers (age unit terminology, see first reviewer comment, it needs to be clear

that the ages in the data files are uncalibrated)

▢ providing a separate column for the type of dated material (wood, bone, bulk peat,

macrofossils, charcoal, etc.) in both files

▢ providing a separate column for the dated archive/basin type (lake, mire or pond)

▢ adding URLs to the radiocarbon reports of the lake/mire isolation data, where available (like

in the archaeological data file).

Although there is a lot of dry work involved, I strongly suggest adding this information to raise the

quality of the data and increase the impact of this publication. If changes were already made and the

changes are just not visible because the files are in review on the PANGAEA website, just ascertain

again, that the adjustments were correct.

Please find a few minor comments and technical issues below. Please excuse me that I might have

overseen some of them in the first review:

We thank Referee #2 again for the constructive comments. The previous comments and the comments after the first revision have improved the quality of the manuscript significantly.

The updated data sets have been uploaded to PANGAEA and the data sets are now visible there. All the requests have been addressed. Here are detailed responses for the other comments. At the end of the document is the marked-up version of the manuscript.

l. 17: Instead of 'timing', the term 'chronological order' should be used.

l. 21: comma after satellites

Both were corrected as suggested.

l. 4: 'on the bottom sediments' instead of 'in'

Corrected as suggested.

l. 5: to the 'time period of Ancylus Lake', the time period should be given in numbers, with a reference.

Time period in numbers and two references were added to the text.

l. 6: isolation and contact contradict each other. You could remove 'contact'.

Corrected as suggested.

ll. 7-8: '…in mire studies and references…', technically, these mire studies are among the references. You can write that there are interpretations of the retreat drawn from mire studies and then put a few literature references behind, which are specifically evidencing it. In other words, if you already use 'references', then also name a few.

Corrected as suggested with references.

l. 10: comma after 'In Fennoscandia'

Corrected as suggested.

l. 13: 'data based on expert judgement on the sediment layers of mires…' sounds weird, as if it is unclear what "expert judgement" is or who these experts are. It would sound more serious if you just

write: 'data based on the initial layers of mire development as well as …'

Corrected as suggested.

ll. 20-21: put 'method' behind 14C.

Corrected as suggested.

'Each data point has a spatial location, elevation and age information that is preferably obtained using the 14C dating and calibration.' Because the chapter is called data description, the sentence suggests or may be misinterpreted, that the ages provided in the files are calibrated already. It should be made clear that the data user still needs to calibrate the 14C-dates. The (raw) 14C-dates are already there, but the calibration still needs to be done, which is not differentiated here yet.

The need to calibrate the 14C dates was emphasized.

l. 3: It is not completely clear what is meant by the statement that 'in some cases the layers of different datings are close to one another in the core.' Does it mean that there are big age discrepancies between samples that are almost in the same layer/depth or very close to each other?

That was the intention and it has been clarified.

ll. 3-4: You could give some hint to the reader for possible reasons for these age reversals. For example, roots that were accidentally dated and give ages that are less old than a layer above, redeposited older material from outside the catchment…. Possibly with a few references. At least, remove the 'for some unknown reasons'. Otherwise, it may sound slightly unprofessional.

An example with a reference was added as well as possible reasons for these age reversals.

In Figure 2, the figure caption still says height instead of elevation. Consider using m a.s.l. (above sea level).

The figure and the figure caption were corrected to use elevation m a.s.l.

ll. 20-21: Again the pointer to the two datasets and the need to apply the changes concerning terminology, units and additional columns on sample type, links to the reports where missing, etc. (see first review).

The PANGAEA data sets have been updated addressing these remarks.

l. 25: Put citation/syntax in order/rephrase to: 'In earlier studies (e.g. Pohjola et al., 2014, 2018), the calibration…..'

Corrected as suggested.

Page 6 (mind that the line numbers are not displayed correctly)

l. 33: on the Estonian side

Corrected as suggested.

ll. 1-5: To streamline the conclusion slightly, consider ending the sentence after the Church et al. reference and then: 'In contrast, the land uplift in the Gulf of Finland is 0-5 mm / year (Poutanen et al., 2010). Therefore, future housing development planning in the northern Baltic Sea coastal areas can benefit from this data set as a tool to assess the flooding probability.'

Corrected as suggested.

Page 8 (References)

ll. 2-3: DOI is missing for this reference.

DOI was added for the reference

All pages:

'modeling' is American English. Change to British 'modelling'. Check for consistency throughout the manuscript.

Page 1, lines 7, 17, 20, Page 2, l. 26., Page 4, l. 10, Page 5, l. 18

Corrected throughout the manuscript.

[revised manuscript text omitted]

---

## Author Response (AR3)

We thank the Topical Editor for these comments regarding minor technical corrections. Here are detailed responses for these comments. At the end of the document is the marked-up version of the manuscript.

Data:

1. Table: Fennoscandia_lake_mire_isolation.tab, rows 261-266 (Hedenström. Early Holocene shore displacement in eastern Svealand, Sweden): The provided link to the pdf (https://pubs.sub.su.se/5454.pdf) is not accessible from outside Sweden. This is ok as most of the data are in the table. However, it would be helpful to add a note in the data table saying "not accessible outside Sweden) and preferably changing the link from "https://pubs.sub.su.se/5454.pdf" to "http://su.diva-portal.org/smash/record.jsf?pid=diva2%3A1094181&dswid=863". The latter represents the landing page of this work and includes the information about the access restriction.

The link was changed by recommendation of PANGAEA to "http://urn.kb.se/resolve?urn=urn:nbn:se:su:diva-143015", which uses the urn to resolve the link to the document. In addition, to the following column (original reference) " ( download not accessible outside Sweden)" behind the reference was added.

2. The DOI of the data are not registered. Please ask PANGAEA to register the DOI after addressing my first point. After the DOI is registered, please change all https://doi.pangaea.de links in the paper to https://doi.org (in the abstract, data availability statement and references)

The DOI of the data has been registered: https://doi.org/10.1594/PANGAEA.905352 and the links in the paper have been changed.

Paper:

• Please address the changes in the DOI link to the data as mentioned above
The changes have been made.
• Page 3, line 6: please change "An example of this kind age reversal..." to "An example for this kind of age reversal..."
Corrected as suggested.

• Page 4, first paragraph, second line: please change "younger than than" to "younger than"

Corrected as suggested.

[revised manuscript text omitted]